# Inhibition of the Low Molecular Weight Protein Tyrosine Phosphatase (LMPTP) as a Potential Therapeutic Strategy for Hepatic Progenitor Cells Lipotoxicity—Short Communication

**DOI:** 10.3390/ijms20235873

**Published:** 2019-11-22

**Authors:** Michalina Alicka, Katarzyna Kornicka-Garbowska, Michael Roecken, Krzysztof Marycz

**Affiliations:** 1Department of Experimental Biology, Wroclaw University of Environmental and Life Sciences, Norwida 27B, 50-375 Wrocław, Poland; michalina.alicka@upwr.edu.pl (M.A.); kornicka.katarzyna@gmail.com (K.K.-G.); 2International Institute of Translational Medicine, Jesionowa, 11, Malin, 55-114 Wisznia Mała, Poland; Michael.Roecken@vetmed.uni-giessen.de; 3Faculty of Veterinary Medicine, Equine Clinic–Equine Surgery, Justus-Liebig-University, 35392 Gieβen, Germany

**Keywords:** LMPTP, LMPTP inhibitor, equine hepatic progenitor-like cells, chaperone-mediated autophagy, mitophagy, endoplasmic reticulum stress

## Abstract

Equine metabolic syndrome (EMS) is a cluster of metabolic disorders, such as obesity, hyperinsulinemia, and hyperleptinemia, as well as insulin resistance (IR). In accordance with the theory linking obesity and IR, excessive accumulation of lipids in insulin-sensitive tissues (lipotoxicity), like liver, alters several cellular functions, including insulin signaling. Therefore, the purpose of the study was to isolate equine hepatic progenitor-like cells (HPCs) and assess whether inhibition of low molecular weight protein tyrosine phosphatase (LMPTP) affects the expression of genes involved in macroautophagy, chaperone-mediated autophagy (CMA), endoplasmic reticulum stress, and mitochondrial dynamics in a palmitate-induced IR model. We demonstrated that LMPTP inhibition significantly enhanced expression of heat shock cognate 70 kDa protein (*HSC70),* lysosome-associated membrane protein 2 (*LAMP2*), and parkin *(PRKN)*, all master regulators of selective autophagy. We also observed downregulation of C/EBP homologous protein *(CHOP),* activating transcription factor 6 *(ATF6)* and binding immunoglobulin protein encoded by the *HSPA* gene. Moreover, LMPTP inhibition increased alternative splicing of X-box binding protein 1 (*XBP1)*, suggesting high endonuclease activity of inositol-requiring enzyme 1 alpha (IRE1α). Taken together, our data provide convincing evidence that LMPTP inhibition reverses palmitate-induced insulin resistance and lipotoxicity. In conclusion, this study highlights the role of LMPTP in the regulation of CMA, mitophagy, and ER stress, and provides a new in vitro model for studying HPC lipotoxicity in pre-clinical research.

## 1. Introduction

Equine metabolic syndrome (EMS) is a set of characteristic symptoms, including insulin resistance (IR), hyperinsulinemia, hyperleptinemia, increased generalized and regional adiposity, as well as local and systemic inflammation [1]. In addition, those metabolic disorders are highly associated with the occurrence of laminitis—an extremely painful and life-threating disease, which can ultimately lead to an animal’s euthanasia. The proper diagnosis and management of EMS is still problematic and represents a real challenge for veterinary medicine. There is evidence that in horses, excessive intake of fodder rich in non-structural carbohydrates (NSCs) increases the activity of hepatic de novo lipogenesis, leading to lipotoxicity, and finally IR [2]. In human, excessive accumulation of toxic triglyceride in the liver cells contributes to the development of non-alcoholic fatty liver disease (NAFLD) [3]. IR occurs primarily in insulin-sensitive tissues, like liver, adipose tissue, and muscle, and may be caused by several mechanisms, e.g., endoplasmic reticulum (ER) stress, impaired mitochondrial dynamics, and autophagy [4,5,6,7].

The insulin receptor (INSR) belongs to the large class of tyrosine kinase receptors. INSR activation initiates the cascade of phosphorylation events, triggering a network of intracellular signaling pathways leading to glucose uptake [8,9]. The application of pharmaceutical products that enhance cellular response to insulin by increasing INSR phosphorylation, and therefore mitigate ER stress and improve mitochondria dynamics, is considered a promising strategy to reduce IR in horses suffering from EMS. Recent studies reported that protein tyrosine phosphatases (PTPs) are involved in insulin signaling regulation [10]. Thus, PTPs may become a useful target in searching for potential drugs in IR treatment. One of the PTPs family member is low molecular weight protein tyrosine phosphatase (LMPTP), encoded by the *ACP1* gene [11]. Human genetic evidence suggests that LMPTP is a key negative regulator in IR and other obesity-related metabolic disorders. LMPTP dephosphorylates and inactivates the INSR, which promotes IR and type 2 diabetes [12,13]. What is more, Pandey et al. revealed that partial knock-down of LMPTP expression improved lipid and glycemic profiles, as well as IR in diet-induced obese C57BL/6 mice [14]. Recently, Stanford et al. synthetized the orally bioavailable, effective LMPTP inhibitor (N,N-diethyl-4-(4-((3-(piperidin-1-yl)propyl)amino)quinolin-2-yl) benzamide), which they entitled “compound 23”. Furthermore, the performed experiments revealed that global and liver-specific LMPTP depletion protects mice from diabetes induced by a high-fat diet without changes in body weight [8]. In addition, Stanford et al. demonstrated that LMPTP plays a critical role in the promotion of adipogenesis [15].

Therefore, in this study, we hypothesized, that inhibition of the LMPTP with compound 23 would improve mitochondrial dynamics and mitigate endoplasmic reticulum stress in palmitate-induced insulin-resistant equine hepatic progenitor-like cells (HPCs). We believe that lipid overload not only in hepatocytes but also in HPCs promotes EMS development by deterioration of the HPC regenerative potential.

## 2. Materials and Methods

All cell culture reagents and chemicals were purchased from Sigma Aldrich (St Louis, MO, USA), unless stated otherwise.

### 2.1. Cell Isolation and Culture

HPCs isolation was performed in accordance with the protocol described previously [16]. In order to isolate the cells, samples of liver tissue derived from non-EMS horses were washed twice with Dulbecco’s phosphate-buffered saline (DPBS) supplemented with 1% antibiotic (penicillin/streptomycin; PS) and chopped into smaller fragments. After mechanical disintegration, specimens were incubated in 0.10% *w*/*v* solution of collagenase IV for 45 min in a CO_2_ incubator. Then, the digested tissue was centrifuged at 1500× *g* for 5 min in 4 °C. The pellet was resuspended in Opti-MEM medium (Thermo Fisher Scientific, Waltham, MA, USA) supplemented with 10% fetal bovine serum (FBS) and 1% PS (complete medium) and seeded in a 6-well plate. Cell immortalization was performed when cells reached 30% of confluence using Lenti-SV40 - tsA58 temperature sensitive mutant (Cat.# LV629, abm, BC, Canada) in accordance with the manufacturer’s protocol. Briefly, cells were rinsed with DPBS and incubated with 1 mL of viral supernatant and polybrene infection/transfection reagent (8 µg/mL). Cells were incubated for 24 h in a CO_2_ incubator (37 °C). Next, media with lentivirus was discarded and cells were washed twice with DPBS. The culture was maintained in complete Opti-MEM medium at 32 °C and 5% CO_2_. Immortalized cells were generated by serial cell passaging that allowed the elimination of non-immortalized cells.

### 2.2. In Vitro Study

Cells were passaged 10 times before the experiment. Liver cells were divided in three groups: Control (CTRL—non-treated cells), treated with palmitate (PA), and treated with palmitate and then with low molecular weight protein tyrosine phosphatase inhibitor (PA_LMPTP_). Briefly, cells were seeded in a 24-well plate at a density of 3 × 10^4^ cells/well in a final volume of 500 µL in complete Opti-MEM medium and left to attach overnight. Then, culture medium was replaced with Opti-MEM supplemented with 1% PS. Incubation was carried out for 2 h in the CO_2_ incubator at 37 °C. Sodium palmitate solution (final concentration 0.6 mM in bovine serum albumin (BSA) solution) was prepared as described previously [17]. The incubation with palmitate was maintained at 37 °C for 24 h. Next, LMPTP inhibitor was added directly to the cell culture medium of PA_LMPTP_ (final concentration: 1 µM) for 6 h. Cell viability was evaluated by staining with calcein-AM and propidium iodine (PI) dyes. The staining was performed in accordance with the manufacturer’s protocol. Viable cells were labelled with calcein-AM and emitted green fluorescence, whereas the nuclei of dead cells with PI emitted red fluorescence. The cells were imaged using an epifluorescent microscope (Axio Observer A1, Zeiss) and captured with a Canon PowerShot camera (Canon, Tokyo, Japan).

### 2.3. RNA Isolation and Gene Expression Profiling Analysis

Total RNA was extracted from cells using TRI Reagent in accordance with the phenol-chloroform RNA isolation protocol [18]. RNA concentration and purity were measured spectrophotometrically (Epoch, Biotek, Germany). Genome DNA was digested with a DNase I RNase-free Kit (Thermo Fisher Scientific, Waltham, MA, USA), and cDNA was synthesized using a Tetro cDNA Synthesis Kit (Bioline, London UK) and T100 Thermal Cycler (Bio-Rad). In total, 150 ng of total RNA were used for each reaction. All qPCR reaction mixtures contained 500 nM of each specific primer, 5 µL of SensiFast SYBR Green Kit (Bioline, London, UK), and 1 µL of cDNA in a final volume of 10 µL. RT-qPCR reactions were performed using the CFX Connect^TM^ Real-Time PCR Detection System (Bio-Rad, CA, USA). The primer sequences are listed in Table 1. The values of the transcripts were normalized in relation to the expression of the glyceraldehyde 3-phosphate dehydrogenase (*GAPDH*) and β-actin (*ACTB*) using the 2^-ΔΔ*C*T^ method by comparing the tested groups to the non-treated control group.

Expressions of *HNF4A, PECAM1, AFP, ALAT, CD90, CD105*, and *EPCAM* were estimated by standard quality RT-PCR using agarose electrophoresis. Similarly, the splicing of *XBP1* was detected using primers designed previously by Cassimeris et al. [19]. The PCRs products were run in 2% gel and separated to visualize a 26-bp shift.

### 2.4. Statistical Analysis

Results are presented as mean ± SD. Statistical comparison between groups was conducted using the one-way ANOVA (and nonparametric) test, followed by Tukey’s test for post-hoc comparison. Each RT-qPCR result was normalized to the non-treated group as the control. Differences were considered statistically significant at: * *p* < 0.05, ** *p* < 0.01, and *** *p* < 0.001.

## 3. Results

### 3.1. HPC Genotype and Morphology

Hepatic progenitor cells (HPCs) are bipotential cells residing in normal liver tissue. They exhibit properties that are intermediate between those of stem cells and mature functional hepatic cells and/or cholangiocytes [16,20]. HPCs were isolated using the enzymatic method and immortalized when they reached 30% confluency with Lenti-SV40 - tsA58 temperature sensitive mutant. Immortalized cells were cultured in Opti-MEM medium supplemented with 10% FBS. In order to determine the HPC population, we compared the corresponding mRNA expression profile of genes encoding (1) HPCs and hepatic specific markers, such as α-fetoprotein (*AFP*), hepatocyte nuclear factor 4 α (*HNF4A*), and alanine aminotransferase (*ALAT*) [21]; (2) HPCs and cholangiocyte-specific markers (*SOX9*) [22] and epithelial cell adhesion molecule (*EPCAM*) [16]; (3) adipose-derived mesenchymal stem cell (ASCs) markers, such as *CD105* and *CD90* [23]; as well as an endothelial cell specific marker, platelet and endothelial cell adhesion molecule 1 (*PECAM1*) [24]. We found that similarly to whole liver tissue, cultured HPCs expressed *HNF4A* and *ALAT.* Moreover, HPCs expressed markers of hepatic progenitor cells, such as *EPCAM* and *SOX9*. On the other hand, ASCs markers, like CD90 and CD105, were not detected in HPCs. Similarly, we did not observe any expression of *AFP* and *PECAM1* (Figure 1A). The results suggest that cultured HPCs expressed markers characteristic for hepatic stem cells, whereas they did not express *AFP* (hepatocytes marker), *PECAM1* (endothelial cells marker), *CD90*, and *CD105* (ASCs markers). Cell morphology was visualized using confocal and bright-field (BF) microscopy. Mitochondria distribution (Figure 1B) and cellular appearance in BF images (Figure 1C) showed that most cultured cells were oval cells with enlarged round-shaped nucleus and low nuclear:cytoplasmic ratio. Similar to our results, some studies reported that histologically, HPCs adopt the appearance of epithelial cells with a scan cytoplasm and oval nucleus [20].

### 3.2. Expression of Apoptosis-Related Genes

To examine the effect of LMPTP inhibition on apoptosis in PA-induced lipotoxicity in HPCs, we treated those cells with 0.6 µM sodium palmitate and LMPTP inhibitor. Previous studies have shown that PA promotes apoptosis in myoblastic cells and human hepatocytes (HepG2) [25,26], thus we decided to analyze the expression of apoptosis-related genes in HPCs from each experimental group using RT-qPCR. We observed no significant differences in the transcript levels of pro-apoptotic *P53* and *BAX*, as well as anti-apoptotic *BCl-2.* Conversely, the *BCl-2/BAX* ratio was significantly increased in the presence of PA (*p* < 0.05) (Figure 2A). In order to confirm the results, cells were subjected to calcein and propidium iodide (PI) staining. We observed a significant increase in PI fluorescence intensity in the PA group. What is more, LMPTP inhibition significantly improved cell viability after PA treatment (Figure 2B). The obtained results suggest that LPTP inhibition may improve cell viability in PA-treated HPCs.

### 3.3. LMPTP Inhibition Alters Expression of Genes Involved in Autophagy, Mitochondria Dynamics, and Unfolded Protein Response (UPR) in IR-Induced HPCs

Lipotoxicity and IR in insulin-sensitive tissues correspond with impaired autophagy and mitochondrial dynamics, as well as excessive ER stress. Therefore, we sought to identify the role of LMPTP inhibition on the expression of autophagy-related genes using RT-qPCR. We observed that LMPTP inhibition significantly reduced the mRNA level of *BECN1* (*p* < 0.001) (Figure 3A), whereas it did not affect expression of *SQSTM1* (Figure 3B) and *LC3* (Figure 3C) compared to the PA group. In addition, it markedly enhanced the expression of genes involved in chaperone-mediated autophagy (CMA), such as *HSC70* (Figure 3D) and *LAMP2* (Figure 3E), in comparison with the PA-induced IR group (*p* < 0.05, *p* < 0.01 and *p* < 0.01, respectively). The upregulation of autophagy-related genes suggests that LMPTP inhibition may help to re-establish cellular homeostasis. An imbalance in the mitochondria fusion/fission ratio is highly linked to mitochondria dysfunction and IR [6]. We found that LMPTP inhibition enhanced mRNA levels of *FIS1* (*p* < 0.05) (Figure 3G); however, it did not alter the expression of *MFN1* (Figure 3F) or the *MFN1/FIS1* ratio (Figure 3H) when compared to the PA group. Mitofusin encoded by the *MFN1* gene promotes mitochondrial fusion, whereas mitochondrial fission 1 (*FIS1*) is involved in mitochondria fission [27]. Importantly, LMPTP inhibitor promoted expression of *PRKN* (Figure 3J)*,* whereas it did not affect the expression of *PINK1* (Figure 3I). Both genes positively regulate mitophagy, which is involved in selective removal of damaged mitochondria and thus participates in mitochondria quality control [28]. In contrast, the result showed that LMPTP inhibitor induced a significant decrease of *PGC1A* transcript levels in comparison to the CTRL and PA groups (*p* < 0.01 and *p* < 0.001, respectively) (Figure 3K). In addition, we examined the effect of LMPTP inhibition on the expression of genes involved in the unfolded protein response (UPR). UPR is a regulatory system activated in response to excessive accumulation of unfolded or misfolded proteins in the ER lumen, while its prolonged activation promotes cell apoptosis [29]. We found that LMPTP inhibition diminished expression of *CHOP* (Figure 3M), *ATF6* (Figure 3N), and *HSPA5* (Figure 3O) when compared to the PA group (*p* < 0.001). What is more, it did not affect the transcript levels of *PERK* (Figure 3L) and *IRE1A* (Figure 3P), whereas it downregulated XBP1 splicing (Figure 3R). RT-PCR analysis showed that 24 h of exposure of HPCs to PA significantly enhanced the levels of the spliced form of x-box binding protein (*sXBP1*) when compared to the control untreated group (*p* < 0.001). In contrast, LMPTP inhibition significantly reduced transcript levels of *sXBP1* (255 bp) (*p* < 0.001) (Figure 3R). Decreased splicing of *XBP1* after LMPTP inhibition indicates increased IRE1α endonuclease activity. The upregulation of mitophagy-related genes along with downregulation of UPR-related genes observed during LMPTP inhibition indicates its role in the modulation and maintenance of metabolic homeostasis.

## 4. Discussion

A growing prevalence of metabolic disorders in human and domestic animals has become an urgent issue in global healthcare. Humans suffering from prolonged IR substantially develop type two diabetes while horses suffer from EMS. Diagnosed animals are characterized by obesity and impaired cellular response to insulin. Lack of proper management leads to disease progression, which may culminate in the occurrence of life-threating laminitis. A large body of evidence indicates that in both horses and humans, metabolic disorders are usually accompanied by lipotoxicity in insulin-sensitive tissues, including liver. Excessive accumulation of lipids in cells drives IR progression, disturbing energy and tissue homeostasis. Thus, therapeutic strategies targeting liver lipotoxicity may become an effece alternative approach for metabolic disorders.

HPCs, small and oval bipotential cells, represent a heterogenous population of cells, which participate in restoration of the damaged liver tissue [30]. Therefore, we decided to isolate HPCs from non-EMS horses, and treated them with PA in order to induce lipotoxicity and IR in the cells. Then, we examined whether LMPTP inhibition was able to reverse the harmful effects of PA treatment. The isolated heterogenous population of cells comprised of a large number of oval cells characterized by enlarged oval nuclei and a high nuclear to cytoplasm ratio. Moreover, isolated cells expressed hepatocytes markers, such as *ALP* and *HNF4A*, and cholangiocytes markers, like *EPCAM* and *SOX9*, which are also characteristic of pluripotent stem cells. Several studies showed that HPCs are known to be clonogenic epithelial cells expressing markers of both hepatocyte and cholangiocyte cell lineages [20,30,31,32]. Moreover, it has been suggested that under conditions of massive liver cell loss, HPCs differentiate into new hepatocytes and/or cholangiocytes [30,31].

One of the most promising targets in slowing down IR development is LMPTP. Several data obtained from human and mouse genetic studies indicate that LMPTP inhibition may reduce the progression of diabetes and IR in obesity. Low LMPTP enzymatic activity protects against hyperlipidemia in obese patients and is associated with reduced glycemic levels in both diabetic and non-diabetic individuals [8,12,13,33]. Moreover, knockdown of LMPTP improves the glycemic profile, diminishes IR, and meliorates INSR phosphorylation in adipocytes and hepatocytes in high-fat diet-induced obese (DIO) C57BL/6 (B6) mice [14]. Thus, we used a novel LMPTP inhibitor (N,N-diethyl-4-(4-((3-(piperidin-1-yl)propyl)amino)quinolin-2-yl) synthesized by Stanford et al. [8] in order to reverse PA-induced changes in the expression of genes involved in cell apoptosis, mitochondrial dynamics, autophagy, and UPR. A growing body of evidence indicates that, in the course of NAFLD, prolonged hepatocyte apoptosis can trigger the proliferation and activation of HPCs [30,34]. Our data revealed that PA, the most common saturated fatty acid, did not affect the expression of pro-apoptotic genes, such as *P53* and *BAX*, whereas it increased the *BCl-2/BAX* ratio and induced cell death, as shown by calcein-AM and PI staining. Follis et al. reported that the imbalance of *BAX* and *BCl-2* expression contributes to apoptosis of hepatocytes in the course of NAFLD. *BAX* is a transcriptional target of pro-apoptotic *P53* [35] and regulates the expression of genes associated with lipid metabolism [36]. What is more, PA decreased mRNA levels of genes involved in CMA, like *HSC70* and *LAMP2*. Additionally, we found that the expression of *PRKN* was significantly reduced after PA treatment. We also observed upregulation of genes involved in UPR, except *IRE1A*. Nevertheless, a significant increase in *XBP1* splicing suggests high endonuclease activity of IRE1α. In contrast, LMPTP inhibition significantly suppressed the negative effects of PA on the expression of genes involved in CMA, mitophagy, and ER stress activation. Several studies revealed that PA induces lipotoxicity and IR in hepatocytes, causing impairment of crucial cellular processes, such as autophagy and mitochondrial dynamics [17,37,38]. CMA, a selective form of autophagy, stimulates lipid droplets’ breakdown to initiate lipolysis via either macroautophagy or cytosolic lipases. Several pieces of evidence suggest that lipid droplets are selectively recognized by the macroautophagic machinery and are effectively incorporated into autophagosomes (double-membrane vesicles formed during the autophagy process) to release free fatty acids (lipophagy) [39,40]. In addition, disturbances in CMA lead to hepatic glycogen depletion and hepatosteatosis, and cause important alternations in lipid and glucose metabolism, as well as in overall organism energetics [41,42]. Dysfunction of CMA in humans is linked to the development of several pathological conditions, such as neurodegenerative diseases, diabetic nephropathy, and fatty liver [40]. Data obtained in presented research strongly suggest that macroautophagy and CMA are involved in the attenuation of lipotoxicity in liver cells. Moreover, the accumulation of saturated fatty acids in the liver may also contribute to ER stress and induce UPR. In general, UPR stands as a protective mechanism, but when prolonged and uncontrolled it contributes to hepatocyte death [29,43,44]. The UPR encompasses three signaling branches named IRE1α, PERK, and ATF6. PERK induces activity of pro-apoptotic transcriptional factor CHOP [45], which in turn inhibits transcription of anti-apoptotic *BCl-2* [46], leading to cell death. On the contrary, ATF6 positively regulates *XBP1* mRNA levels, the spliced form of which activates the UPR efficiently [47]. IRE1α has the ability to splice XBP1 mRNA (*sXBP1*), which after translocation to the nucleus, initiates the expression of genes involved in the regulation of ER stress [45]. Furthermore, increased levels of *sXBP1* significantly enhance the expression of genes involved in lipogenesis, such as sterol regulatory element-binding protein (*SREBP-1*)c and fatty acid synthase (*FASN*), exacerbating lipid accumulation in the liver [48,49].

Taken together, in this study, we investigated the effects of LMPTP inhibition in a PA-induced HPCs model. Our data revealed a partial reversal of the negative effects of PA treatment at the molecular level. Therefore, we conclude that LMPTP may be a promising candidate for the treatment of EMS, particularly in horses suffering from liver lipotoxicity. By enhancing CMA and mitophagy while diminishing ER stress, it may improve insulin sensitivity and inhibit disease progression. Thus, further in vitro and in vivo research is desirable in order to fully understand the molecular mechanism in which LMPTP sensitizes liver cells to insulin.

## Figures and Tables

**Figure 1 ijms-20-05873-f001:**
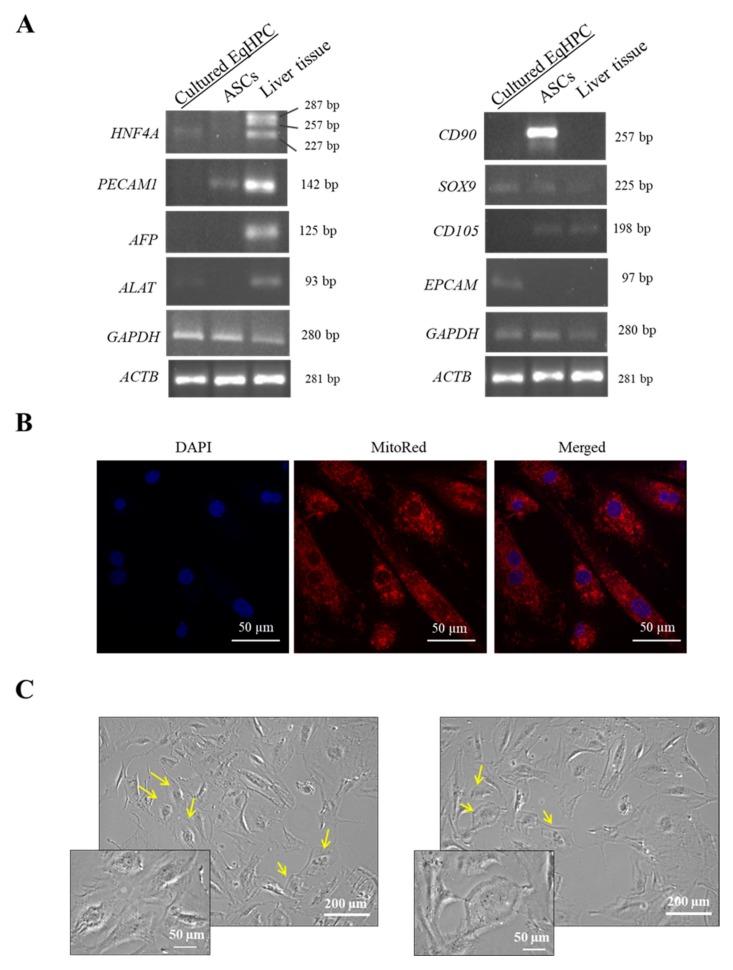
Genotype and morphology of cultured HPCs. (**A**) Phenotype of HPCs was determined using RT-PCR. The PCR products were run in 2% agarose gel. Both *GAPDH* and *ACTB* were used as housekeeping genes. (**B**) Representative confocal microscopy of mitochondria staining with MitoRed. (**C**) The morphology of cultured HPCs. Yellow arrows indicate the typical appearance of oval cells.

**Figure 2 ijms-20-05873-f002:**
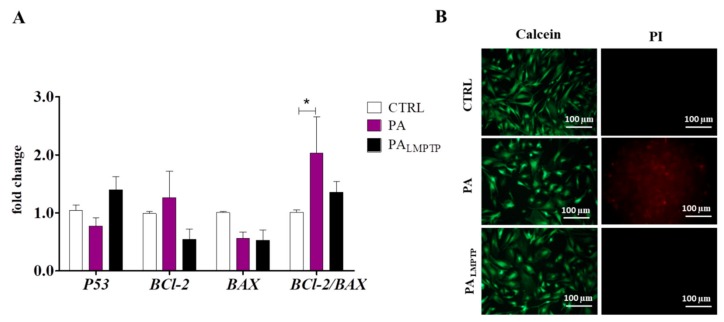
Determination of cell apoptosis using RT-qPCR (**A**) and calceinacetoxymethyl (AM) / propidium iodine (PI) double staining (**B**). Apoptosis was evaluated by analysis of apoptosis-related genes expression. Results are expressed as mean ± SD. Statistical significance is indicated as an asterisk (*): * *p* < 0.05, using a one-way ANOVA (and nonparametric) test. (**B**) Representative images of calcein-AM (green) and PI (red) staining. The images were taken using an epifluorescence microscope.

**Figure 3 ijms-20-05873-f003:**
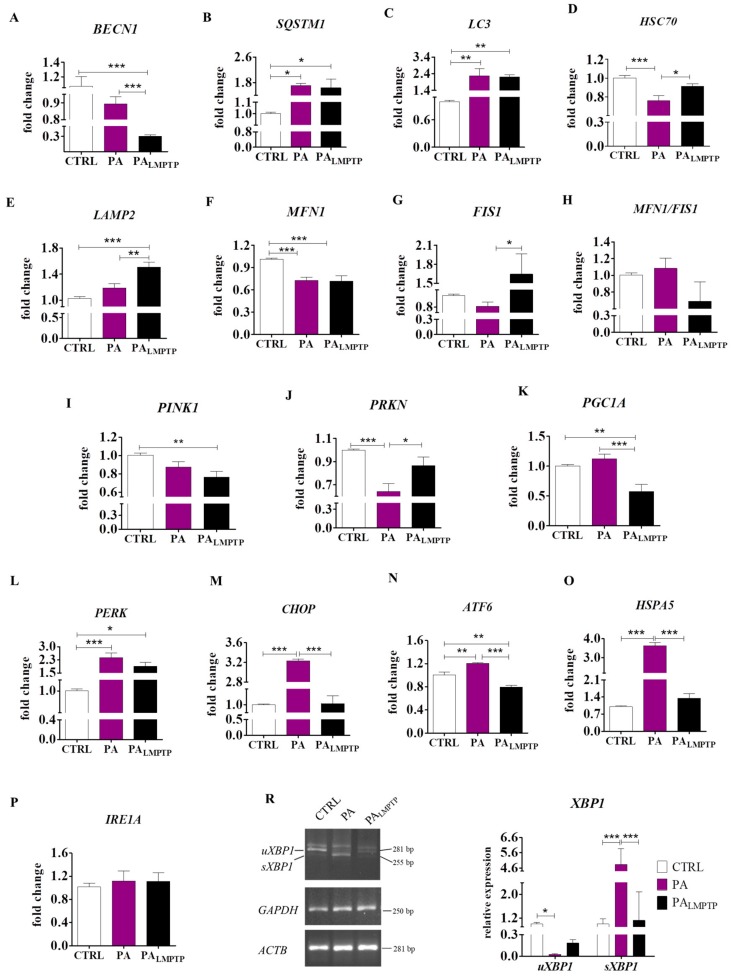
The effect of low molecular weight protein tyrosine phosphatase (LMPTP) inhibition on the expression of genes involved in the regulation of autophagy (**A**–**E**), mitochondrial dynamics (**F**–**K**), and unfolded protein response (UPR) (**L**–**R**). (**A**–**P**) Transcript levels of investigated genes were assed using RT-qPCR. (R) In order to determine mRNA levels of x-box binding protein (*uXBP1*) and *sXBP1*, T-PCR was performed. The PCR products were run in 2% agarose gel. Relative quantities of *uXBP1* and *sXBP1* were evaluated using Image Lab software after normalization with both glyceraldehyde 3-phosphate dehydrogenase (*GAPDH*) and β-actin (*ACTB*) as housekeeping genes. Results are expressed as mean ± SD. Statistical significance is indicated as an asterisk (*): * *p* < 0.05, ** *p* < 0.01, *** *p* < 0.001 using the one-way ANOVA (and nonparametric) test.

**Table 1 ijms-20-05873-t001:** Sequences of primers used in RT-qPCR.

Gene	Primer Sequence 5′→3′	Amplicon Length (bp)	Accession No.
*HNF4A*	F: CAGGAGATGCTGCTGGGAG R: ATTGTGGTGATGGCTCCTGG	257	XM_014735168.2
*PECAM1*	F: ACACGGAAGTGGAAGTGACC R: CCATCAAGGGAGCCTTCCG	142	NM_001101655.2
*AFP*	F: CAGCCACTTGTTGCCAACTC R: CTGGCCAACACCAGGGTTTA	125	NM_001081952.1
*ALAT*	F: GGGAAGGCACCTACCACTTC R: ACTTGGCATGGAACTGGCTT	93	XM_005613421.2
*CD90*	F: CTCCCACCCCTGGTGAAAACT R: CGGTGGTATTCTCATGGCGG	257	XM_001503225.4
*SOX9*	F: GAACGCCTTCATGGTGTGGG R: TTCTTCACCGACTTCCTCCG	225	XM_014736619.1
*CD105*	F: GACTGCCTTTGTGCAGTTGG R: ATGCTTTCGGGGTCCTTCAG	198	XM_003364144.4
*EPCAM*	F: TTGCCGTCATTGTGGTTGTG R: TCAGCCTTCTCGTACTTCGC	97	XM_001917795.4
*P53*	F: TACTCCCCTGCCCTCAACAA R: AGGAATCAGGGCCTTGAGGA	252	U37120.1
*BCl-2*	F: TTCTTTGAGTTCGGTGGGGT R: GGGCCGTACAGTTCCACAA	164	XM_014843802.1
*BAX*	F: CGAGTGGCAGCTGAGATGTT R: AAGGAAGTCCAGTGTCCAGC	153	XM_023650077.1
*BECN1*	F: GATGCGTTATGCCCAGATGC R: ATCCAGCGAACACTCTTGGG	147	XM_014729146.1
*SQSTM1*	F: CATCGGAGGATCCCAGTGTG R: CCGGTTTGTTAGGGTCGGAA	207	XM_005599173.3
*LC3*	F: TTCTGAGACACAGTCGGAGC R: CTTTGTTCGAAGGTGTGGCG	128	XM_001493613.6
*HSC70*	F: GATTAACAAGAGGGCTGTCCGTC R: GCCTGGGTGCTAGAAGAGAGA	122	XM_023628864.1
*LAMP2*	F: GCACCCCTGGGAAGTTCTTA R: ATCCAGCGAACACTCTTGGG	147	XM_014831347.1
*MFN1*	F: AAGTGGCATTTTTCGGCAGG R: TCCATATGAAGGGCATGGGC	217	XM_014838357.1
*FIS1*	F: GGTGCGAAGCAAGTACAACG R: GTTGCCCACAGCCAGATAGA	118	XM_014854003.1
*PINK1*	F: GCACAATGAGCCAGGAGCTA R: GGGGTATTCACGCGAAGGTA	298	XM_014737247.1
*PRKN*	F: TCCCAGTGGAGGTCGATTCT R: CCCTCCAGGTGTGTTCGTTT	218	XM_014858374.1
*PGC1A*	F: TCTACCTAGGATGCATGG R: GTGCAAGTAGAAACACTGC	93	XM_005608845.2
*PERK*	F: GTGACTGCAATGGACCAGGA R: TCACGTGCTCACGAGGATATT	283	XM_023618757.1
*CHOP*	F: AGCCAAAATCAGAGCCGGAA R: GGGGTCAAGAGTGGTGAAGG	272	XM_001488999.4
*ATF6*	F: CAGGGTGCACTAGAACAGGG R: AATGTGTCTCCCCTTCTGCG	164	XM_023640315.1
*HSPA5*	F: CTGTAGCGTATGGTGCTGCT R: CATGACACCTCCCACGGTTT	122	XM_023628864.1
*IRE1A*	F: GAATCAGACGAGCACCCGAA R: TTTCTTGCAGAGGCCGAAGT	300	XM_023652216.1
*XBP1*	F: TTACGCGAGAAAACTCATGGCC R: GGGTCCAAGTTGAACAGAATGC	281 (unspliced) 255 (spliced)	XM_014742035.2
*GAPDH*	F: GATGCCCCAATGTTTGTGA R: AAGCAGGGATGATGTTCTGG	250	NM 001163856.1
*ACTB*	F: GATGATGATATCGCCGCGCTC R: CGCAGCTCGTTGTAGAAGGT	281	NM_001081838.1

Sequence, product size, and accession numbers of the primer set. *HNF4A*: hepatocyte nuclear factor 4 alpha; *PECAM1:* platelet endothelial cell adhesion molecule; *AFP*: alpha-fetoprotein; *ALAT*: alanine aminotransferase; *CD73:* ecto-5’-nucleotidase; *CD90*: Thy-1, cluster of differentiation 90; *SOX9*: transcription factor SOX9; *CD105*: endoglin, ENG; *EPCAM*: epithelial cell adhesion molecule, *P53*: tumor suppressor p53; *BCl-2:* B-cell lymphoma 2; *BAX*: BCl-2 associated X protein; *BECN1*: beclin1; *SQSTM1*: sequestosome; *LC3*: microtubule-associated proteins 1A/1B light chain 3B; HSC70: heat shock cognate; *LAMP2:* lysosome-associated membrane protein 2; *MFN1*: mitofusin 1; *FIS1*: mitochondrial fission 1 molecule; *PINK1*: PTEN-induced putative kinase 1 (PINK1); *PRKN:* parkin RBR E3 ubiquitin protein ligase (PARK2); *PGC1A:* peroxisome proliferator activated receptor gamma coactivator 1 alpha; *PERK*: protein kinase RNA-like endoplasmic reticulum kinase; *ATF6*: activating transcription factor 6; *HSPA5*: binding immunoglobulin protein; *IRE1A*:inositol-requiring enzyme 1 alpha; *XBP1*: X-box binding protein 1; *GADPH*: glyceraldehyde-3-phosphate dehydrogenase, *ACTB:* beta actin.

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
