# Peer review of "Inhibition of the Low Molecular Weight Protein Tyrosine Phosphatase (LMPTP) as a Potential Therapeutic Strategy for Hepatic Progenitor Cells Lipotoxicity—Short Communication"

_ijms, 2019, doi:10.3390/ijms20235873_

Round 1
Reviewer 1 Report
The reviewed paper is very interesting. However, based on my previous experience, the expression level of GAPDH (glyceraldehyde‐3‐phosphate dehydrogenase) gene is quite unstable in several cell lines. Therefore, the Authors should replace this reference gene. For example, some genes encoding tubulin or actin may be applied instead. In addition, I suggest to improve Discussion section in order to perform in-depth interpretation of the valuable results.
Author Response
We would kindly thank to the reviewer for careful reading of this manuscript, meaningful suggestions and valuable comments. Based on these comments and suggestions, we have performed careful modifications to the original manuscript. The point-to-point responses are shown below.
Reviewer 1
The reviewed paper is very interesting. However, based on my previous experience, the expression level of GAPDH (glyceraldehyde‐3‐phosphate dehydrogenase) gene is quite unstable in several cell lines. Therefore, the Authors should replace this reference gene. For example, some genes encoding tubulin or actin may be applied instead. In addition, I suggest to improve Discussion section in order to perform in-depth interpretation of the valuable results.
Response: We performed additional RT-PCRs and RT-qPCRs to improve the results using ACTNB (coded β-actin) as reference gene. Interestingly, we have observed changes in expressions of P53, BCL-2, BAX, SQSTM1, LC3 and PINK1. We have merged previous and current results to obtain reliable statistical data. A summary of the results are presented in the manuscript, whereas the results compared to GAPDH (Figure 1) and ACTNB (Figure 2) are set of below (please find the attachment).
Discussion section was improved.
Reviewer 2 Report
In the presented article the authors evaluate the inhibition of LMPTP as potential therapeutic strategy for hepatic progenitor cells lipotoxicity. Authors carried about careful experiments and got related results, so the manner of scientific work is satisfactory. The manuscript is not easy to read and not so well understand in several places. In my opinion, to make it easy to understand for readers, authors should revise the manuscript to improve logical flow, transitions between the sections, and language.
My specific comments are as follows:
Point 1. line 27 page 1, the authors need to standardised style and font in the abstract.
Point 2. line 37 and line 42 page 1, these sentences should be rearranged in different way because some words is missing and sentences are confusing.
Point 3. line 113 page 3, usually abbreviation for Quantitative reverse transcription PCR is RT-qPCR. Should be change everywhere in the text.
Point 4. line 164 page 6, in the Figure 1 Phenotype should be changed for genotype because the authors investigated in this case gene expression and not protein expression of used genes.
Point 5: line 183 page 7, in the Figure 2 the relative expression should be change in fold change if the authors used deltadeltaCt methods or relative expression of gene of interest (2-deltaCt).
Point 6: line 222 page 9, The Figure 3 have the same objection as for figure 2. The authors should change the graphs as indicated previously.
Point 7. In the discussion part of the article the style and font need to standardise.
Point 8. line 276 page 10, the sentence should be rearranged in different way because some words is missing and the meaning is not sound reasonable.
There are several errors of English grammar and typographical mistakes throughout the manuscript, and these should be corrected.
Author Response
We are very grateful to the reviewers for their thoughtful suggestions and critical comments. All of the notes help us to improve the quality of this article. The point-to-point replies and explanations for all of the suggestions are listed below.
Reviewer 2
In the presented article the authors evaluate the inhibition of LMPTP as potential therapeutic strategy for hepatic progenitor cells lipotoxicity. Authors carried about careful experiments and got related results, so the manner of scientific work is satisfactory. The manuscript is not easy to read and not so well understand in several places. In my opinion, to make it easy to understand for readers, authors should revise the manuscript to improve logical flow, transitions between the sections, and language.
My specific comments are as follows:
Point 1. line 27 page 1, the authors need to standardised style and font in the abstract.
Response to point 1: The style and fond have been standarised.
Point 2. line 37 and line 42 page 1, these sentences should be rearranged in different way because some words is missing and sentences are confusing.
Response to point 2: The sentences have been corrected.
Point 3. line 113 page 3, usually abbreviation for Quantitative reverse transcription PCR is RT-qPCR. Should be change everywhere in the text.
Response to point 3: The qRT-PCR abbreviation has been replaced with RT-qPCR.
Point 4. line 164 page 6, in the Figure 1 Phenotype should be changed for genotype because the authors investigated in this case gene expression and not protein expression of used genes.
Response to point 4: The phenotype has been replaced with genotype.
Point 5. line 183 page 7, in the Figure 2 the relative expression should be change in fold change if the authors used deltadeltaCt methods or relative expression of gene of interest (2-deltaCt).
Response point 5: The relative expression have been changed in fold change.
Point 6. line 222 page 9, The Figure 3 have the same objection as for figure 2. The authors should change the graphs as indicated previously.
Response point 6: Corrected.
Point 7. In the discussion part of the article the style and font need to standardise.
Response to point 7: Corrected.
Point 8. line 276 page 10, the sentence should be rearranged in different way because some words is missing and the meaning is not sound reasonable.
Response to point 8: The sentence has been rearranged.
There are several errors of English grammar and typographical mistakes throughout the manuscript, and these should be corrected.
The errors in English grammar and typographical mistakes have been corrected.
Round 2
Reviewer 1 Report
The Autors significantly improved the manuscipt, therefore, I recommend publishing it in IJMS journal. I suggest minor corrections of English style and grammar.
Author Response
We are very grateful to the reviewers for their thoughtful suggestions and critical comments. All of the notes help us to improve the quality of this article. The response is showed below:
Reviewer 1
The Autors significantly improved the manuscipt, therefore, I recommend publishing it in IJMS journal. I suggest minor corrections of English style and grammar.
Response: The English grammar corrections have been performed.
Reviewer 2 Report
Dear authors,
in order to improve the quality of your paper within mine suggestion you have done a good job.
The points about the possible mechanisms of action of Inhibition of the low molecular weight protein tyrosine phosphatase (LMPTP) as a potential therapeutic strategy for hepatic progenitor cells lipotoxicity are still very general and haven't been address any possible mechanism of action. Readers get little insight into properties of the therapeutic candidate.
Regarding Fig .3; is still incomplete and remains difficult to understand. On to x axis I suggest to put the legend because one graph has it and other are without.
Some parts of the manuscript remain difficult to understand in particular the discussion part need to be more "strained forward" emphasized conclusions.
Author Response
We would kindly thank to the reviewer for careful reading of this manuscript, meaningful suggestions and valuable comments. Based on these comments and suggestions, we have performed careful modifications to the original manuscript. The point-to-point responses are shown below.
Reviewer 2
The points about the possible mechanisms of action of Inhibition of the low molecular weight protein tyrosine phosphatase (LMPTP) as a potential therapeutic strategy for hepatic progenitor cells lipotoxicity are still very general and haven't been address any possible mechanism of action. Readers get little insight into properties of the therapeutic candidate.
Response: The possible mechanisms of action of inhibition of the low molecular weight protein tyrosine phosphatase (LMPTP) is described in more detail. Whereas the role of LMPTP inhibition in the liver is still not well understood.
Regarding Fig .3; is still incomplete and remains difficult to understand. On to x axis I suggest to put the legend because one graph has it and other are without.
Response: The Figure 3 legend has been corrected.
Some parts of the manuscript remain difficult to understand in particular the discussion part need to be more "strained forward" emphasized conclusions.
Response: The discussion has been corrected.